# Effect of Pulsed Electric Fields on the Lipidomic Profile of Lipid Extracted from Hoki Fish Male Gonad

**DOI:** 10.3390/foods11040610

**Published:** 2022-02-21

**Authors:** Anna Burnett, Mirja Kaizer Ahmmed, Alan Carne, Hong (Sabrina) Tian, Isam A. Mohamed Ahmed, Fahad Y. Al-Juhaimi, Alaa El-Din Ahmed Bekhit

**Affiliations:** 1Department of Food Sciences, University of Otago, P.O. Box 56, Dunedin 9054, New Zealand; buran002@student.otago.ac.nz (A.B.); mirja.ahmmed@postgrad.otago.ac.nz (M.K.A.); 2Department of Fishing and Post-harvest Technology, Faculty of Fisheries, Chittagong Veterinary and Animal Sciences University, Bangladesh, Khulshi, Chittagong 4225, Bangladesh; 3Department of Biochemistry, University of Otago, P.O. Box 56, Dunedin 9054, New Zealand; alan.carne@otago.ac.nz; 4Sanford Limited, 22 Jellicoe Street, Auckland 1010, New Zealand; stian@sanford.co.nz; 5Department of Food Science and Nutrition, College of Food and Agricultural Sciences, King Saud University 2460, Riyadh 11451, Saudi Arabia; iali@ksu.edu.sa (I.A.M.A.); faljuhaimi@ksu.edu.sa (F.Y.A.-J.)

**Keywords:** hoki male gonad, lipid extraction, PEF, non-thermal food processing, fatty acid composition, phospholipids, NMR, ETHEX

## Abstract

Processing of hoki, a commercially important fish species, generates substantial quantities of co-products, including male gonad, which contains valuable lipids, such as phospholipids, that could be recovered and utilised. Hoki fish male gonads (HMG) were subjected to pulsed electric fields (PEF) treatment at varying field strengths (0.625, 1.25, and 1.875 kV/cm) and frequencies (25, 50, and 100 Hz), at a fixed pulse width of 20 μs. The total lipid was extracted using an ethanol-hexane-based (ETHEX) extraction method, and the phospholipid and fatty acid compositions were determined using ^31^P NMR and GC-FID, respectively. The total lipid yield was increased from 4.1% to 6.7% by a relatively mild PEF pre-treatment at a field strength of 1.25 kV/cm and frequency of 50 Hz. A higher amount of EPA (8.2%), DPA (2.7%), and DHA (35.7%) were obtained by that treatment, compared to both un-heated (EPA: 8%; DPA: 2.5%; DHA: 35.2%) and heat-treated controls (EPA: 7.9%; DPA: 2.5%; DHA: 34%). No significant changes to the content of the major phospholipids were observed. PEF pre-treatment under mild conditions has potential for improving the total lipid yield extracted from fish male gonad.

## 1. Introduction

Fish and marine foods are well known sources of polyunsaturated fatty acids (PUFAs) and phospholipids (PLs) [1]. For example, fish contain high amounts of PUFAs, ranging from 24% to 65% of total lipid, which includes substantial amounts of eicosapentaenoic acid (EPA), docosapentaenoic acid (DPA), and docosahexaenoic acid (DHA) [2]. About 70% of the fish harvested from the World’s oceans and from aquacultures is processed into various forms of products, such as dried, canned, and frozen, and is mainly focused on the fish muscle [3,4]. Fish processing generates substantial amounts of co-products, such as fish frame, trimmings, head, skin, scales, roe (gonad), and viscera [4] that are typically used either as fish meal, biofertilizer, for the production of oil, or are discarded. Some of these co-products, such as head, roe, skin, and gonad are increasingly recognized for their phospholipid content, which comprises substantial amounts of bioactive components such as EPA and DHA [5,6,7,8]. Previous literature has demonstrated the health benefits of n-3 phospholipids, specifically, EPA, DPA, and DHA, that can enhance brain health and create a positive outcome for the prevention of neurodegenerative disorders, cardiovascular disease, non-alcoholic fatty liver disease, diabetes, and some cancers [2].

A few studies have investigated the composition of male gonad of fish. A previous study reported the proximate composition of the male gonad (testes) of several red sea fish species and found some variation in proximate composition (moisture: 72–73%, lipid: 0.9–1.2%, protein: 22–24% and ash: 2.5–2.7%, based on % of wet weight), which also varied depending on season [9]. A recent study reported the proximate composition (moisture: 67%; protein: 67%; lipid: 6.9%; and ash content: 1.2%), fatty acid (total n-3: 42.5%; EPA:12.1%; DPA:1.64%; DHA:24.7%), and phospholipid profile (total phospholipid: 1.86% of wet tissue; major phospholipids: PC, PE, LDPG, and SM) of blue mackerel gonad [6]. A previous study investigated the fatty acid composition in the gonads of several ray species (*Dasyatis marmorata*, *Rhinobatos cemiculus*, and *Rhinoptera marginata*), and reported a higher lipid content, ranging from 4.20 to 5.42% wet weight, in male gonad, compared to female roe, which ranged from 3.30 to 4.96% wet weight [10]. Another study reported a higher lipid content in king angelfish (*Holacanthus passer*) male gonad (ranging from 11.26 to 11.80% wet weight), compared with that found in rays [11]. The lipid content of king anglefish female roe (4.97–6.08% wet weight) was similarly found to be lower than that of the male gonad. Lower lipid contents have been reported for male gonads of *Lethrinus genivittatus* Safaga (1.05% wet weight), *Balistoides viridescens* Safaga (1.21% wet weight), and *Salmo salar* (1.2–2.8% wet weight) [9,12]. To the best of our knowledge, there is no information available on hoki male gonad (HMG), an underutilized co-product that may contain substantial amounts of valuable polyunsaturated fatty acids (PUFAs) and phospholipids (PLs). Hoki is a commercially important fish species with global harvesting output reported to have been in the vicinity of 67–73 million tonnes of wet weight over the 2016–2019 period [13]. The gonad and liver of hoki has been reported to comprise 10–25% of the total fish, and are under-utilized co-products, not considered as a delicacy [5]. However, the utilization of this underutilized co-product may provide economic and environmental benefit to the fish industry.

Pulsed electric fields (PEF) treatment involves the exposure of biological cells to high intensity electric field pulses that can alter the structure of the cell membrane. The external electrical field promotes cell electroporation, causing the cell membrane barrier to be compromised and become permeable. Although PEF is commonly used in industry to inactivate microorganisms and extend the shelf life of food products, research has also demonstrated the capability of PEF treatment to enhance the extraction of valuable compounds from plant and animal tissue. Published studies have highlighted the application of PEF in the extraction of valuable compounds from various marine organisms [14,15,16]. However, research is limited on the use of this technique for the extraction of lipids from marine sources, especially gonads. Therefore, the objective of the present study was to investigate the use of PEF as a pre-treatment for enhancing the extraction of lipid from HMG, and to examine whether PEF has any effect on the composition and quality of the fatty acids and phospholipids in the extracted lipid.

## 2. Materials and Methods

### 2.1. Chemicals and Sample Preparation

Hoki male gonad (HMG, 10 kg) was supplied by Sanford Ltd., a New Zealand commercial fish processing company. The HMG was received in March 2021 and stored in sub-aliquots at −20 °C until required for processing. Gonad material was thawed in a chiller room at 4 °C for 12 h, and the weight (34.70 ± 10.12 g), length (10.92 ± 2.06 cm) and width (2.54 ± 0.66 cm) of the individual gonads were recorded prior to processing. Three individual 1 kg samples of pooled gonads were used as sample replicates for analyses. Twenty subsamples of 50 g each (produced from each of the three 1 kg pooled gonad samples) were vacuum packed and stored at −20 °C for further analysis. Nine subsamples from each of the three batches were subjected to PEF treatment at varying frequencies of 25, 50, or 100 Hz and field strengths of 0.625, 1.25, or 1.875 kV/cm, with a fixed pulse width of 20 µs. Each subsample was subjected to one set of parameters and each treatment was conducted in three replications. The PEF processing conditions were chosen based on published literature [17,18,19], along with running preliminary trials in order to observe the response of the material prior to a wide range of treatment conditions to establish suitable working conditions, in relation to avoiding flashover (electrical arcing) and excessive heating. The PEF system used was an Elcrack-HPV5 (DIL, Quakenbruck, Germany) that contained a power generator, a treatment chamber, and an oscilloscope (Model UT2025C, Uni-Trend Group Ltd., Hong Kong, China), in order to monitor the square wave bipolar pulse shape used during treatment. A batch process was used during operation of the PEF system and the treatment chamber used had a triangle shaped compartment of dimensions 6 × 6 × 4 cm. The distance between the two electrodes was 4 cm, and electrodes were insulated with Teflon insulating material. The weight of each sample (g) was measured prior to each PEF treatment. The temperature and conductivity of each sample was measured using a conductometer (CyberScan CON 11, Eutech Instruments, Singapore), before and after the PEF treatment. Three gonad samples were retained as replicate un-treated controls, and three additional samples were heat-treated at 45 °C for 1 min using a water bath (Techne, TE-10A Tempette, Total Lab Systems Ltd., Auckland, New Zealand) to act as heat-treated controls, representing the maximum temperature that resulted in PEF treatment of the male gonad material. The processed male gonad samples were stored in airtight bags at −20 °C for further processing.

### 2.2. Measurements

#### 2.2.1. Electrical Input

Values from electrical treatment parameters (pulsed electric field strength, pulse peak voltage, peak current, peak power, pulsed energy, energy amount, pulse count and flashover) were recorded from the PEF system for each treatment. The specific energy input Ws (kJ/kg) was calculated according to the following equation [20]:Ws = E2 × τρ × σ × n(1)
where E is the electric field strength, τρ is the pulse width, σ is the conductivity, and n is the number of pulses.

#### 2.2.2. Temperature and Electrical Conductivity

The electrical conductivity (mS·cm^−1^) and the temperature (°C) of the HMG were measured immediately before and after PEF treatment using a conductometer electrode (CyberScan CON 11, Eutech Instruments, Singapore).

### 2.3. Proximate Analysis

Proximate analysis of HMG samples was carried out using standard AOAC methods (AOAC, 1990) to determine the percentage moisture, ash, and total protein. The lipid content was determined using the “ETHEX” method [5], as described in Section 2.4. Carbohydrate was determined from the sum of the moisture, lipid, protein, and ash % subtracted from 100%. The results are reported on both a wet and dry weight basis to facilitate comparison with the literature in which results are presented on both a wet weight and dry weight basis.

### 2.4. Total Lipid Extraction

The recently published ETHEX method [5] is described as a fast and effective lipid extraction method that generates a higher lipid yield compared to other common lipid extraction methods such as the FOLCH [21] and Bligh and Dyer methods [22]. Lipid extraction was performed on ground wet HMG using ethanol (95% *v*/*v*) and hexane (99% *v*/*v*) solvents in a 2:1 ratio. Ground wet HMG (30 g) was added to a Schott bottle along with 300 mL of ethanol/hexane solvent and was then homogenized on ice for 5 min at 15,000 rpm, using an Ultraturrax (IKA-Werke GmbH and Co. KG, Stufen, Germany). The mixture was then centrifuged for 15 min at 1900× *g* in a J2-21M/E centrifuge at −5 °C (Beckman Coulter, Inc., Fullerton, CA, USA), and then filtered under gentle suction through Whatman no.1 filter paper. The container was washed twice with the same solvents and then filtered again. Following this, the filtrate was transferred into a round-bottomed flask, and a rotary evaporator (Buchi, Flawil, Switzerland) was used to evaporate the solvent.

### 2.5. Fatty Acid Methyl Ester (FAME) Analysis by GC-FID

The extracted lipid samples were used for the preparation of fatty acid methyl esters (FAME) that were then analyzed using gas chromatography-flame ionization detection (GC-FID).

#### 2.5.1. Sample Preparation

Each lipid sample (20 mg) was dissolved in 1 mL of chloroform/methanol solution before adding 3 mL of 0.5 M KOH in methanol. The mixture was then vortexed and heated for 20 min at 80 °C. After cooling for 5 min, 2 mL of diethyl ether and 5 mL of Milli-Q-water were added, mixed by gentle rocking, and left to settle for 5 min to allow a clarified diethyl ether upper layer to form. Using a glass Pasteur pipette, the upper layer was discarded and the bottom phase was acidified with concentrated hydrochloric acid until blue litmus indicator paper turned red. Then, 2 mL of diethyl ether was added and the mixture was mixed by gentle rocking, and then it was left to phase separate. The upper layer was then transferred to a KIMAX glass tube and 1 mL of 14% (*v*/*v*) boron trifluoride (BF3) in methanol was added. The mixture was then heated at 80 °C for 20 min. Once cooled to room temperature, 5 mL of saturated NaCl solution was added and the mixture was left to phase separate for 5 min. The upper organic layer was transferred to a 2 mL GC vial for further analysis.

#### 2.5.2. GC-FID Operating Conditions

FAME samples were analyzed by injecting 1 µL of the FAME sample into the GC-FID. Hydrogen was used as the carrier gas at a flow rate of 1 mL/min and a split ratio of 30:1. FAME separation was achieved on a BPX-70 (0.32 mm; film thickness 0.25 µm, length 50 m) silica column (Phenomenex, Torrance, CA, USA). The initial column temperature was 100 °C, which was increased to 160 °C at a rate of 10 °C/min, and then up to 220 °C at a rate of 3 °C/min. This temperature was maintained for 5 min, and then increased to 260 °C at a rate of 10 ºC/min, and then held for a further 5 min. The retention times of 37 fatty acid methyl esters were determined using commercial FAME standards (FAMQ-005, AccuStandard, Inc., New Haven, CT, USA), which were analysed by GC-FID under identical running conditions. Docosapentaenoic acid (DPA) and several unknown peaks were identified based on the reported literature [5,6,7,8].

### 2.6. Phospholipid Analysis Using ^31^P NMR

The ^31^P NMR analysis was conducted using a Varian MR 400 NMR (Agilent Technologies, Santa Clara, CA, USA.) [5]. The analysis was conducted under the following running conditions: buffer pH, 7.4; temperature, 25 °C; scan number, 192; relaxation delay, 3.5 s; sweep width, 6067 Hz; data points, 65,536; and excitation pulse, 90 °C, along with proton decoupling. PC at −0.87 ppm was used to reference the chemical shift of each phospholipid compound. In order to reference and identify the phospholipid compounds, glyphosate (1 µmol/mL) was used as an internal standard. Commercial pure standards were used to determine the chemical shift of each PL and the quantification of the phospholipids was determined as described previously [5].

### 2.7. Statistical Analysis

The experiments were replicated (*n* = 3), and the results were subjected to analysis of variance (ANOVA) using general linear model (GLM) protocol in Minitab, version 16.1. (Minitab Limited, Sydney, Australia). The model investigated the effects of PEF field strength (kV/cm), frequency (Hz), and their interactions on the measured parameters. The homogeneity of variance and normality of the data were determined using Bartlett’s test and Shapiro–Wilk test, respectively. The results for the heated and un-treated controls were analyzed using one-way ANOVA to investigate the effect of heating on the measured parameters. To separate the means, Tukey’s test at 0.05 level of significance was used and the results are reported as means ± standard error of the mean.

## 3. Results and Discussion

### 3.1. Proximate Analysis of HMG

Table 1 shows the proximate composition of HMG on a wet weight and dry weight basis, as the values reported in the literature are a mixture of being based on wet weight or on dry weight. The moisture content of HMG was found to be higher than the literature values reported for hoki roe [5,23], male gonad of Pacific blue mackerel, *Lethrinus genivittatus* Safaga, *Balistoides viridescens* Safaga, and king anglefish, and ranged between 67.2 to 73.4% [6,9], but was similar to that reported for Atlantic salmon (80–82%) [11].

It has been previously reported that there is seasonal variation in the proximate composition of hoki, where moisture content of the white muscle of male hoki ranged from 79.5 to 86.6% and the total protein content ranged from 11.4 to 18.5%, with higher values found in fish not in the spawning season [24]. Migration or spawning causes protein to be metabolised for gonad development and energy, which can also lead to a significant reduction in the protein content of hoki muscle tissue [24]. This can explain the differences observed between the protein content of HMG (11.77%, Table 1) and the protein content of male gonad of Pacific blue mackerel (23.1 ± 3.0%), *Lethrinus genivittatus* Safaga (23.56%), and *Balistoides viridescens* Safaga (23.77%) [6,9]. The total lipid content of male gonad was reported for carp (*Cyprinus carpio*) to be 2.49 ± 0.17 [25], 1.63–1.79% in *Zoarces viviparus* [26], 2.1% in cod [27], and 2.16–2.64% in Skipjack tuna [28]. These results are lower than that for HMG obtained in the present study. Similarly, a slightly lower lipid content (19.1% on a dry weight basis) was reported for *Inimicus japonicus* [29]. The ash content of HMG was similar to that reported for blue mackerel gonad, king angelfish gonad, hoki roe, and Southern blue whiting roe reported earlier [6,11,23].

### 3.2. Effect of Pulsed Electric Fields on Hoki Roe

Hoki male gonad samples were subjected to PEF treatment under various conditions prior to extraction of the lipid. With increasing PEF field strength and frequency, the joule heat generation increased and the highest temperature (45 °C) was found at the highest PEF field strength of 1.875 kV and frequency of 100 Hz that was used in the study (Table 2).

Figure 1 shows the effect of PEF pre-treatment on the total lipid yield extracted from the HMG samples. Overall, there was an increase in lipid yield following PEF treatment, but the only significant increase (*p* < 0.05) was achieved with PEF pre-treatment at a field strength of 1.25 kV/cm and frequency of 50 Hz (Figure 1). This indicates that the application of these PEF conditions was sufficient to cause structural modification in the male gonad tissue, which enhanced the subsequent extraction of lipid, in that PEF mediated electroporation resulted in enhanced permeabilization of cell membranes, allowing for greater movement of solvent into the sample during lipid extraction [30]. However, the use of PEF as a pre-treatment did not result in a significant (*p* > 0.05) difference in total lipid yield, compared to that obtained from un-treated HMG. Previous studies on the application of PEF in the fish processing industry have reported positive results for the technology. For example, Zhou et al. (2017) found that PEF can enhance extraction and protein yield from mussels [19]. Li et al. (2016) found an enhanced extraction yield of abalone viscera protein from *Haliotis discus hannai ino* viscera through the application of PEF [31]. Similarly, He et al. (2017) demonstrated the ability of PEF to enhance the extraction and yield of chondroitin sulfate, calcium, and collagen from fish bone [18]. It is not understood as to why relatively mild PEF conditions led to an enhanced lipid yield from HMG, in the present study (Figure 1), however, the milder PEF conditions would result in less heating of the male gonad material, which may have an effect on lipid extraction. Although previous studies found increased extractability of protein, calcium, chondroitin sulfate, and collagen from fish bone of various marine species, which correlated with higher PEF intensity, HMG may respond differently to PEF treatment, and further investigation is required to understand the mechanisms involved [18,19,31].

### 3.3. Effect of Heat Generated by PEF Treatment on the Composition of Lipid Extracted from HMG

#### 3.3.1. Effect of Heat on Fatty Acid Composition

To investigate whether heat had any effect on the lipid yield and lipid characteristics, three samples were subjected to heating at 45 °C (the highest temperature found during any of the PEF pre-treatments of HMG) using a water bath. Heat-treated controls enable assessment of whether any change to lipid characteristics following PEF pre-treatment is due to the heating or due to the PEF treatment itself. The effect of heat on the fatty acid composition of the total lipid extracted from HMG is outlined in Table 3. Some significant differences (*p* < 0.05) were observed between the fatty acid composition of the un-heated controls and the heat-treated controls. Specifically, a significant difference (*p* < 0.05) was found between the amount of C16:0 and the total SFA present in the un-heated control, compared to the heat-treated control. A 2.1% increase in total SFA was found following heat treatment of HMG samples compared to the un-heated controls (Table 3). This result is consistent with a previous study reported earlier [32] that found when heating oils, trans fatty acid (TFA) and SFA content increases due to oxidation. This increase in total SFA in the heat-treated sample was paralleled by an increase in C16:0, as mentioned above. The increase in fatty acids was not limited to SFA, as several MUFA were also increased (C15:1; C16:1 n7; C18:1 n9 cis; and C20:1). In addition, there was a decrease in some SFA (C24:0), MUFA (C14:1 and C17:1), and PUFA (C18:3 n6 and C20:3 n3), as a result of the heating effect. The long chain saturated and unsaturated fatty acids (both MUFA and PUFA) could be decomposed and oxidized at high temperature and could convert into secondary oxidative products [33,34,35], resulting in reduction of the fatty acid content. This phenomenon might explain the apparent decrease in the amount of SFA (C24:0), MUFA (C14:1 and C17:1), and PUFA (C18:3 n6 and C20:3 n3) in the heat-treated control.

#### 3.3.2. Effect of Heat on the Phospholipid Composition

To observe the effect of the heat generated by PEF on the phospholipid composition, HMG samples were heat-treated in a water bath at 45 °C and the phospholipid composition of extracted lipid compared with that of un-treated controls. The effect of heat on the phospholipid distribution (µmol/g wet tissue) of lipid extracted from HMG is presented in Table 4. Some significant (*p* < 0.05) changes were observed between the un-heated controls and heat-treated controls. Specifically, cardiolipin [23], N-acyl phosphatidylethanolamine (NPE), lyso-phosphatidylethanolamine (LPE), lyso-phosphatidylcholine-1 (LPC-1), and lyso-phosphatidylcholine-2 (LPC-2) were found to decrease significantly (*p* < 0.05) as a result of the heat treatment, demonstrating that heat treatment affected the extracted phospholipid composition of the male gonad samples.

### 3.4. Effect of PEF Treatment on the Lipid Composition of HMG

#### 3.4.1. Effect on Fatty Acids

##### Effect of Frequency on Fatty Acid Composition

Table 5 summarizes the fatty acid composition of the total lipid extracted from HMG pre-treated with PEF at varying treatment intensities. As expected, the results show some significant (*p* < 0.05) variations in fatty acid composition between the 25 Hz and 100 Hz frequency treatments. A significant (*p* < 0.05) increase was found in the C16:0; C18:0; and C18:1 n9 cis content, which is consistent with the effect observed due to heating for C16:0 and C18:1 n9 cis. Increasing the PEF frequency decreased the levels of SFA (C14:0; C22:0; and C23:0), MUFA (C16:1 n7; C17:1; C20:1; C22:1; and C24:1) and PUFA (C18:3 n3; C20:2 n6; and C20:4 n6) fatty acids. Although the overall MUFA and PUFA content remained similar across all three PEF frequencies tested, however, the SFA content was found to increase slightly with the increase in frequency from 25 Hz to 100 Hz during PEF treatment (Table 5). The DHA content was also found to increase with increasing PEF frequency, although EPA and DPA contents remained similar across all PEF treatments (Table 5). No significant (*p* > 0.05) difference due to PEF frequency was observed with the n-3 % content in lipid extracted from the HMG samples. The n-6 % in samples treated with 25 Hz was found to be higher than in those treated with 50 Hz and 100 Hz (*p* < 0.05).

##### Effect of PEF Intensity on Fatty Acid Composition

Table 5 indicates that there were significant variations (*p* < 0.05) in most of the fatty acid compositions in relation to PEF field strength (0.625, 1.25, and 1.875 kV/cm). The highest PEF field strength (1.875 kV/cm) was found to cause the highest reduction in C14-C18 fatty acids, except for C 17:1 (Table 5). The lowest PEF field strength (0.625 kV/cm) was found to decrease the content of C14:0; C15:1; C16:1 n7; C18:3 n3; C20:1; C20:3 n3; and C20:5 n3, compared with the 1.25 kV/cm PEF field strength. MUFA and SFA were significantly higher (*p* < 0.05) at lower PEF, with lower field strengths of 0.625, and 1.25 kV/cm. However, PUFA, was found to be significantly higher (*p* < 0.05) at the highest PEF intensity of 1.875 kV/cm that was investigated. The yield of EPA, DPA, and DHA, along with the n-3 and n-6 contents, were also found to be higher in the HMG samples treated with PEF intensity 1.875 kV/cm. In addition to lower PEF pre-treatment intensities producing higher MUFA and SFA levels in extracted lipid, the elaidic acid (C18:1 n9 trans) and palmitic acid (C16:0) content in extracted lipid was also found to be higher in samples pre-treated with lower PEF intensities. Additionally, DPA was found to be significantly higher (*p* < 0.05) in lipid extracted following PEF pre-treatment with a field strength of 1.875 kV/cm (Table 5). It is well known that PEF treatment causes electroporation of cell membranes [14] and facilitates the release of cellular contents. The above results suggest that different fatty acids respond differently to the PEF treatment intensity.

##### Effect of the Interaction between Frequency and PEF Field Strength on Fatty Acid Composition of Extracted Lipid

The fatty acid composition of the total lipid extracted from HMG pre-treated with PEF at various conditions (frequency: 25, 50, and 100 Hz; PEF field strength: 0.625, 1.25, and 1.875 kV/cm) is summarized in Table 5. The % DPA and SFA contents were found to be most affected by the interaction between frequency and PEF field strength (*p* < 0.05). A PEF field strength of 1.875 kV/cm, in combination with each of the three frequencies investigated, had the greatest effect on DPA content across the hoki gonad samples (Table 5).

The results show that the MUFA content ranged from 19.2 to 20.7%, the PUFA content ranged from 48.9 to 52.9%, and the SFA content ranged from 24.6 to 27.8%. The EPA content was found to range from 7.61% to 8.34%, and the DHA content from 33.8 to 36.2%. As seen in Table 5, these results are consistent with that of the un-treated control and heat-treated control, demonstrating that PEF treatment had no effect on the fatty acid composition of the total lipid extracted from the HMG samples. The overall content of MUFA, PUFA, and SFA in the HMG was found to be similar to that reported for the male gonad of common carp (MUFA: 16.4 ± 5.12, PUFA: 52.2 ± 6.72, SFA: 31.1 ± 2.81) [25]. Suloma and Ogata (2012) also reported similar findings for MUFA (22.34 ± 0.02%) and SFA (30.71 ± 0.96%) in African catfish testis, however, the EPA (2.88 ± 0.12%) and DHA (7.60 ± 0.60%) contents were lower than that found in the present study for HMG (Table 5). Ahmmed et al. (2021) also reported high levels of PUFA (44.7%), along with levels of MUFA (20.7%) and SFA (28.4%) in Pacific blue mackerel gonad [6] that are similar to those reported here in the present study on HMG.

However, the level of EPA (12.1%) found in the male gonad of Pacific blue mackerel was slightly higher than that in HMG (7.61% to 8.34%), whereas DHA (33.8 to 36.2%) was found to be higher in HMG compared to the Pacific blue mackerel gonad (24.7%). C16:0 in HMG (17.3 to 20.9%) was similar to that found in the blue mackerel gonad (19.3%), although C18:1 n9 trans was slightly higher in HMG (12.3 to 12.8%). These slight variations from results presented in the literature could be attributed to differences such as species, seasonality, gender, and age, as previously reported [36]. However, overall the results presented in Table 4 and Table 5 demonstrate that HMG is a good source of PUFA, MUFA, and phospholipid EPA and DHA, of which high dietary intakes have been linked to many health benefits for the human body. Additionally, the ratio of n6/n3 was also found to be less than 1 in lipid extracted from all HMG samples treated with PEF (Table 5). This is important as a balanced omega-6/omega-3 ratio is considered to be beneficial for health and for the prevention of metabolic diseases [37].

#### 3.4.2. Effect on Phospholipid Composition of Lipid Extracted from PEF Pre-Treated HMG

##### Effect of PEF Frequency on Phospholipid Composition

n-3 fatty acids, when esterified to phospholipids, are reported to have increased bioavailability in the brain, as they are able to cross the blood brain barrier [2]. Therefore, it is important to know the phospholipid composition of total lipid extracted from samples such as HMG, in relation to the health-promoting value of the lipid. The ^31^P NMR spectra of the un-heated control, heat-treated control, and PEF-treated sample (50 Hz, 1.25 kV/cm) indicated that these PEF treatment conditions resulted in the highest lipid extraction yield, as presented in Appendix A, and the peak assignment with chemical shift (ppm) is outlined in Appendix A. Significant variations (*p* < 0.05) were found through increasing the PEF frequency from 25 Hz to 100 Hz. This was evident for the LDPG and LPS content, which increased from 1.78 to 2.28% and 1.92 to 2.63%, respectively, with increasing frequency. PC and PE were the most dominant phospholipid classes found in lipid extracted from HMG. This is in agreement with the literature that indicates there are substantial amounts of these phospholipids in fish reproductive organs [6,7,27,35]. In the present study, PC was found to increase with an increase in PEF frequency, however, PE was found to decrease with increasing PEF frequency (Table 6). As mentioned earlier, the electroporation caused by PEF could enhance the release of cellular contents. However, with the increase in treatment intensity, heat generated due to joule effect appeared to degrade some compounds such as PE.

##### Effect of PEF Field Strength on Phospholipid Composition

Significant differences (*p* < 0.05) in the phospholipid composition of HMG samples were obtained through the use of PEF at different field strengths (0.625, 1.25, and 1.875 kV/cm). The LDPG and LPS contents were found to increase with increasing PEF field strength from 0.625 to 1.875 kV/cm (Table 6). A PEF field strength of 1.25 kV/cm resulted in the highest yield of PC (53.3%) in the extracted lipid. However, the PE and PS content decreased with increasing PEF field strength (*p* < 0.05). A PEF field strength of 0.625 kV/cm caused an increase in the yield of PE and PS to 23.7% and 4.99%, respectively. PE and PS are both considered to be essential phospholipids in the human body, participating in many important pathophysiological processes [37]. PS is an acidic phospholipid that can be located on the interior of the cell membrane, allowing for interaction with various cellular components and enhancing cell signalling [38]. PE is also found primarily on the interior of the cell membrane, commonly acting as a ‘chaperone’ to facilitate membrane protein folding, enabling them to function properly [37]. Table 6 presents the phospholipid composition of lipid extracted from HMG (µmol/g wet tissue). The PE content was found to be between 3.30 and 3.81 µmol/g wet tissue. This is substantially higher than that reported by Ahmmed et al. (2021) for the PE content of the male gonad of Pacific blue mackerel, which was found to be 1.05 µmol/g wet tissue [6]. However, Falch et al. (2006) reported the PE content from the milt of spawning cod to be 35.5%, somewhat higher than the 23.7% found for HMG [27]. The composition of PS (4.54–4.99%) in HMG was also found to be similar to that of *Catla catla* roe, which is reported to contain 5.3% PS [38]. PS is reported to be an important bioactive phospholipid for healthy nerve cell membranes and myelin [1]. PS (300–800 mg/d) is reported to be able to be absorbed by human brain efficiently and can potentially reverse biochemical alterations responsible for short-term memory loss [1]. As hoki gonad contains a considerable amount of PS (4.54–4.99%), it could be a good source of bioactive natural PS, which would be beneficial for brain health.

##### Effect of the Interaction between PEF Frequency and PEF Field Strength on Phospholipid Composition

Table 6 highlights the effect of the interaction between frequency and PEF field strength on the phospholipid composition (µmol/g wet tissue) of HMG samples, for which no significant differences were observed. Additionally, the phospholipid composition of the un-heated control and heat-treated control (Table 4) were found to be similar to that of the PEF-treated samples, suggesting that PEF had no effect on the phospholipid composition of lipid extracted from HMG. PC was the main phospholipid present in all samples, followed by PE, cardiolipin [23], and PS. PC is reported to be the most abundant phospholipid in all mammalian cells, comprising between 40–50% of total cellular phospholipid, whereas PE is the second most abundant, comprising between 15–25% of total cellular phospholipid [39,40,41]. The molar ratio of PC and PE is said to be a key determinant of liver health, as changes in the hepatic PC:PE ratio have been linked to the development of non-fatty liver disease (NAFLD) in humans, and furthermore, this ratio is also reported to influence energy metabolism and has been linked to the progression of diseases such as atherosclerosis, insulin resistance, and obesity [39,40,41]. The highest PC content (54.3%) in lipid extracted from HMG in the present study was produced with the interaction of 50 Hz PEF frequency and a PEF field strength of 1.25 kV/cm. A previous study reported a similar result of 55–60% PC content for roe lipid of Catla and *Cirrhinus mrigala* [38]. In addition, a previous study reported that the ovaries and testes of a range fish species have a PC content of 55–80% [42]. However, the PC content of HMG was found to be higher than that for spawning cod milt (43.2%) and Pacific blue mackerel gonad (2.39 µmol/g wet tissue) [6,27].

CL was the third most abundant phospholipid found in the hoki gonad samples. CL is a crucial phospholipid in mammalian cells, playing roles in mitochondrial function and many cellular processes outside of the mitochondria [43]. CL has been reported to have cardioprotective properties in the human body through processes such as mitochondrial bioenergetics, autophagy, and mitogen activated protein kinase pathways [43]. PEF pre-treatment (100 Hz and 1.25 kV/cm) generated the highest amount of CL (5.39%) in lipid extracts of hoki gonad samples. This CL content was found to be substantially higher than that reported in hoki roe lipid, which was found to be 2.7% [44]. Another phospholipid that was detected in substantial amounts was sphingomyelin (SM) [1]. SM plays an important role in roe and gonad through regulating gamete maturation and death [45], which provides a reason as to why it is a substantial component in HMG (Table 6). SM is also crucial in the human diet, especially for infants. A previous study identified a positive association between the neuro-behavioral development of very low birth weight infants and the consumption of SM-fortified milk [46].

## 4. Conclusions

The present study focused on analysing the effect of PEF on the extracted lipid yield and lipid profile of HMG. The fatty acid composition and phospholipid profile of lipid extracted from HMG suggests it can provide a good source of n-3 phospholipids for nutraceutical applications. In the present study, a significant increase (*p* < 0.05) in lipid yield was obtained following mild PEF treatment (1.25 kV/cm and 50 Hz), resulting on average in a 6.81% yield of extracted lipid from HMG. Very little difference was detected in HMG samples pre-treated with PEF in terms of the extracted lipid. The phospholipid composition was not substantially changed, and there was an increase in the level of n-3 fatty acids extracted. The result of this study indicates the potential use of PEF for enhancement of lipid yield extracted from fish gonad, with retention of lipid quality.

## Figures and Tables

**Figure 1 foods-11-00610-f001:**
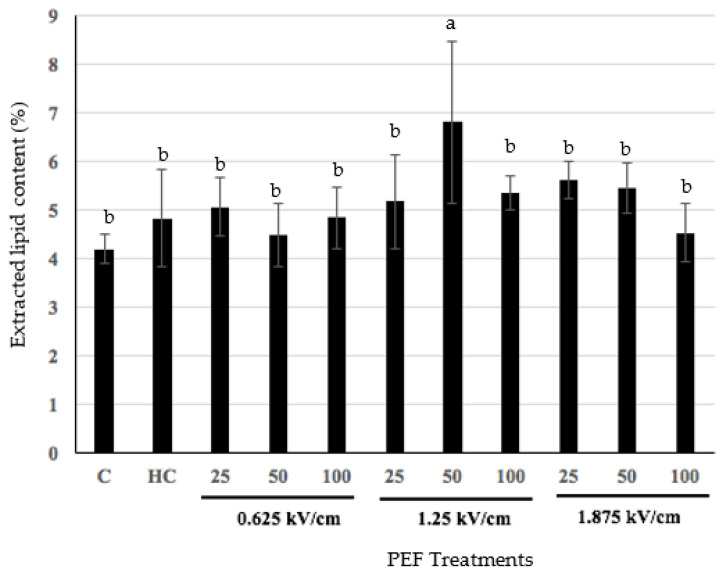
Effect of frequency and pulsed electric field strength on extracted lipid yield (%) from HMG samples. C represents un-heated control, and HC represents heat-treated control. Different letters (a,b) for PEF parameters (frequency and PEF field strength) indicate a significant difference (*p* < 0.05) in lipid yield.

**Table 1 foods-11-00610-t001:** Proximate composition of HMG.

Percentage	HMG (*w/w*)	HMG (*d*/*w*)
Moisture	81.5 ± 0.53	-
Protein ^a^	11.8 ± 0.78	63.8 ± 0.31
Lipid	4.1 ± 0.12	22.2 ± 1.11
Ash	1.3 ± 0.21	7.0 ± 0.13
Carbohydrate ^b^	1.3 ± 0.62	7.0 ± 1.10

Values are presented as % of wet weight (HMG (*w*/*w*)) and % of dry weight (HMG (*d*/*w*)) of HMG. Values are presented as mean ± standard deviation of three individual replications. ^a^ 6.25 was used as the conversion factor to convert total nitrogen to total protein [6,7,8]. ^b^ Carbohydrate was calculated by: 100—sum (% moisture + protein + lipid + ash contents).

**Table 2 foods-11-00610-t002:** PEF pre-treatment conditions and corresponding specific energy inputs.

Treatments	Temperature (°C)	Conductivity (S/m)	Specific Energy Input (kJ/kg)
Electric field strength (kV/cm)			
0.625	15.6 ^c^	16.7	90.2 ^c^
1.25	21.6 ^b^	17.4	222.4 ^b^
1.875	34.1 ^a^	16.9	313.6 ^a^
SEM	1.2	0.6	6.9
*p* value	0.001	0.72	0.001
Frequency (Hz)			
25	18.0 ^c^	17.2	102.2 ^c^
50	23.0 ^b^	16.6	196.2 ^b^
100	30.3 ^a^	17.2	327.9 ^a^
SEM	1.2	0.6	6.9
*p* value	0.001	0.66	0.001
Interaction			
0.625 × 25	15.1 ^d^	17.1	29.5 ^e^
0.625 × 50	16.9 ^d^	16.5	80.7 ^de^
0.625 × 100	17.8 ^cd^	16.6	160.6 ^c^
1.25 × 25	16.5 ^d^	17.9	99.3^d^
1.25 × 50	20.2 ^cd^	16.5	181.3 ^c^
1.25 × 100	28.2 ^bc^	17.8	386.5 ^ab^
1.875 × 25	22.6 ^cd^	16.7	177.7 ^c^
1.875 × 50	34.9 ^ab^	16.9	326.7 ^b^
1.875 × 100	45.0 ^a^	17.2	436.5 ^a^
SEM	2.1	1.03	12.0
*p* value	0.004	0.94	0.001

Means with different superscript letters (a–) in the same column for each PEF field strength, frequency, or their interaction, indicate significant difference (*p* < 0.05). SEM, standard error of mean.

**Table 3 foods-11-00610-t003:** Fatty acid composition of total lipid extracted from HMG.

Fatty Acid	Un-Heated Control	Heat-Treated Control	*p*
C14:0	0.97	1.18	0.16
C14:1	0.07 ^a^	0.00 ^b^	0.001
C15:1	0.28 ^b^	0.32 ^a^	0.01
C16:0	17.7 ^b^	19.5 ^a^	0.001
C16:1 n7	1.97 ^b^	2.26 ^a^	0.02
C17:0	0.30	0.30	0.14
C17:1	0.37 ^a^	0.32 ^b^	0.001
C18:0	4.51	4.52	0.82
C18:1 n9 trans	12.6	13.7	0.11
C18:1 n9 cis	3.71 ^b^	4.15 ^a^	0.04
C18:2 n6 trans	0.17	0.18	0.08
C18:2 n6 cis	0.60	0.64	0.09
C18:3 n6	0.10 ^a^	0.00 ^b^	0.001
C18:3 n3	0.19	0.22	0.12
C20:1	0.20 ^b^	0.26 ^a^	0.03
C20:2 n6	1.98	2.51	0.07
C21:0	0.21	0.22	0.24
C20:3 n3	2.38 ^a^	2.05 ^b^	0.01
C20:4 n6	0.13	0.14	0.39
C22:0	0.86	0.95	0.09
C22:1 n9	0.25	0.22	0.24
C20:5 n3 [EPA]	8.01	7.87	0.54
C23:0	0.12	0.16	0.34
C24:0	0.39 ^a^	0.34 ^b^	0.001
C24:1	0.13	0.18	0.12
C22:5 n3 [DPA]	2.53	2.50	0.57
C22:6 n3 [DHA]	35.2	33.9	0.31
Unknown	4.07	3.32	0.10
MUFA	19.6	21.5	0.06
PUFA	51.3	50.1	0.38
SFA	25.1 ^b^	27.2 ^a^	0.001
n-3	48.3	46.6	0.27
n-6	3.00	3.48	0.12
n6/n3	0.06	0.07	0.12

Values are presented as % of total fatty acids. Un-heated control: samples with no PEF pre-treatment. Heat-treated control: samples heated at 45 °C and no PEF pre-treatment. Data were obtained by GC-FID. Different superscript letters (a and b) in each row indicate significant difference (*p* < 0.05) between heat-treated and un-heated controls. Means that are not statistically different for each fatty acid do not have any superscripts. Abbreviations: MUFA, monounsaturated fatty acid; PUFA, polyunsaturated fatty acid; SFA, saturated fatty acid, EPA, eicosapentaenoic acid; DPA, docosapentaenoic acid; DHA, docosahexaenoic acid.

**Table 4 foods-11-00610-t004:** Extracted phospholipid composition of HMG.

PL	Control	Heat-Treated Control	*p*
LDPG	0.23	0.24	0.87
PG	0.13 ^b^	0.16 ^a^	0.02
CL N-acyl PE	0.86 ^a^ 0.32 ^a^	0.62 ^b^ 0.00 ^b^	0.00 0.00
LPE	0.10 ^a^	0.08 ^b^	0.00
LPSP	0.49	0.46	0.14
LPS	0.19	0.25	0.45
SM	0.38	0.38	0.75
PE	3.90	3.64	0.32
LPC-1	0.57 ^a^	0.33 ^b^	0.00
PS	0.86	0.80	0.07
LPC-2	0.09 ^a^	0.00 ^b^	0.00
PI	0.15	0.16	0.62
PC	7.78	8.09	0.56

Values are presented as µmol/g wet tissue. Un-heated control: samples with no PEF pre-treatment. Heat-treated control: samples heated at 45 °C and no PEF pre-treatment. Data were obtained by ^31^P NMR. Different superscript letters (a and b) in the same row indicate significant difference (*p* < 0.05) between heat-treated and un-heated controls. Means that are not statistically different for each phospholipid do not have any superscripts. Abbreviation: PA = phosphatidic acid; LDPG = lyso-diphosphatidylglycerol; CL = cardiolipin; LPE = lyso-phosphatidylethanolamine; LP S = lyso-phosphatidylserine; SM = sphingomyelin; PE = phosphatidylethanolamine; LPC = lysophosphatidylcholine; PS = phosphatidylserine; PI = phosphatidylinositol; PC = phosphatidylcholine.

**Table 5 foods-11-00610-t005:** Fatty acid composition of total lipid extracted from HMG pre-treated with PEF.

Fatty Acid	PEF Frequency			PEF Field Strength			PEF Interaction										*p*		
	25	50	100	0.625	1.25	1.875	25 × 0.625	25 × 1.25	25 × 1.875	50 × 0.625	50 × 1.25	50 × 1.875	100 × 0.625	100 × 1.25	100 × 1.875	SEM	Freq	PEF	FRE*PEF
C14:0	1.16 ^a^	1.11 ^ab^	1.02 ^b^	1.04 ^b^	1.28 ^a^	0.97 ^b^	1.08 ^cd^	1.37 ^a^	1.02 ^cd^	1.00 ^cd^	1.35 ^ab^	0.99 ^cd^	1.03 ^cd^	1.13 ^bc^	0.89 ^d^	0.03	0.004	0.001	0.13
C15:1	0.30	0.30	0.29	0.29 ^b^	0.33 ^a^	0.27 ^c^	0.29 ^cde^	0.34 ^a^	0.27 ^de^	0.30 ^bcd^	0.33 ^ab^	0.27 ^de^	0.29 ^bcde^	0.31 ^abc^	0.26 ^e^	0.01	0.69	0.001	0.29
C16:0	18.4 ^b^	19.2 ^a^	19.1 ^a^	19.9 ^a^	19.4 ^a^	17.3 ^b^	18.8 ^c^	19.1 ^bc^	17.3 ^d^	20.9 ^a^	19.3 ^bc^	17.4 ^d^	20.0 ^ab^	19.9 ^abc^	17.3 ^d^	0.23	0.002	0.001	0.002
C16:1 n7	2.15 ^a^	2.12 ^a^	2.02 ^b^	2.03 ^b^	2.25 ^a^	2.01 ^b^	2.10 ^c^	2.32 ^a^	2.04 ^c^	1.99 ^c^	2.32 ^ab^	2.04 ^c^	2.00 ^c^	2.11 ^bc^	1.94 ^c^	0.04	0.002	0.001	0.10
C17:0	0.29	0.30	0.30	0.32 ^a^	0.30 ^ab^	0.28 ^b^	0.30 ^ab^	0.30 ^ab^	0.28 ^b^	0.34 ^a^	0.29 ^ab^	0.28 ^b^	0.31 ^ab^	0.30 ^ab^	0.28 ^ab^	0.01	0.67	0.002	0.33
C17:0	0.29	0.29	0.30	0.31 ^a^	0.30 ^a^	0.28 ^b^	0.30 ^abc^	0.30 ^ab^	0.28 ^cd^	0.31 ^ab^	0.29 ^abcd^	0.28 ^d^	0.31 ^a^	0.30 ^abc^	0.28 ^bcd^	0.01	0.29	0.001	0.55
C17:1	0.36 ^a^	0.33 ^ab^	0.30 ^b^	0.31	0.32	0.35	0.35 ^a^	0.37 ^a^	0.35 ^a^	0.30 ^a^	0.32 ^a^	0.37 ^a^	0.29 ^a^	0.27 ^a^	0.34 ^a^	0.02	0.02	0.08	0.42
C18:0	4.40 ^b^	4.43 ^b^	4.53 ^a^	4.54 ^a^	4.44 ^b^	4.38 ^b^	4.44 ^abc^	4.40 ^bc^	4.35 ^c^	4.58 ^ab^	4.38 ^bc^	4.33 ^c^	4.60 ^a^	4.55 ^ab^	4.46 ^abc^	0.04	0.001	0.001	0.22
C18:1 n9 trans	12.8	12.9	12.6	12.8 ^a^	13.1 ^a^	12.4 ^b^	12.8 ^ab^	13.3 ^a^	12.3 ^b^	12.9 ^ab^	13.3 ^a^	12.7 ^ab^	12.8 ^ab^	12.8 ^ab^	12.3 ^b^	0.17	0.11	0.001	0.40
C18:1 n9 cis	3.86 ^b^	4.00 ^a^	4.05 ^a^	4.16 ^a^	4.00 ^b^	3.74 ^c^	3.96 ^bc^	3.89 ^cd^	3.74 ^cd^	4.36 ^a^	3.94 ^bcd^	3.69 ^d^	4.17 ^ab^	4.17 ^ab^	3.80 ^cd^	0.05	0.001	0.001	0.002
C18:3 n3	0.22 ^a^	0.20 ^ab^	0.20 ^b^	0.19 ^b^	0.22 ^a^	0.21 ^ab^	0.20 ^ab^	0.24 ^a^	0.21 ^ab^	0.18 ^b^	0.22 ^ab^	0.21 ^ab^	0.19 ^b^	0.20 ^b^	0.20 ^b^	0.01	0.004	0.01	0.23
C20:1	0.23 ^a^	0.21 ^ab^	0.20 ^b^	0.19 ^b^	0.23 ^a^	0.21 ^a^	0.21 ^abc^	0.25 ^a^	0.22 ^abc^	0.19 ^bc^	0.24 ^ab^	0.22 ^abc^	0.19 ^c^	0.21 ^abc^	0.21 ^abc^	0.01	0.01	0.001	0.54
C20:2 n6	2.32 ^a^	2.17 ^ab^	2.05 ^b^	2.13	2.18	2.23	2.29 ^ab^	2.31 ^ab^	2.38 ^a^	2.04 ^ab^	2.24 ^ab^	2.23 ^ab^	2.07 ^ab^	2.00 ^b^	2.07 ^ab^	0.07	0.001	0.33	0.58
C21:0	0.21	0.20	0.20	0.20	0.21	0.20	0.20	0.22	0.20	0.20	0.21	0.20	0.20	0.20 ^a^	0.20 ^a^	0.01	0.30	0.21	0.24
C20:3 n3	2.11	2.12	2.19	2.06 ^b^	2.13 ^ab^	2.22 ^a^	2.06 ^b^	2.08 ^b^	2.18 ^ab^	2.03 ^b^	2.18 ^ab^	2.16 ^ab^	2.09 ^b^	2.14 ^ab^	2.33 ^a^	0.05	0.07	0.002	0.26
C20:4 n6	0.13 ^b^	0.15 ^a^	0.08 ^c^	0.13 ^a^	0.11 ^b^	0.13 ^a^	0.13 ^b^	0.14 ^b^	0.13 ^b^	0.13 ^b^	0.19 ^a^	0.13 ^b^	0.13 ^b^	0.00 ^c^	0.13 ^b^	0.00	0.001	0.001	0.001
C22:0	0.92 ^a^	0.90 ^ab^	0.87 ^b^	0.85 ^b^	0.90 ^b^	0.94 ^a^	0.91 ^abc^	0.90 ^abc^	0.97 ^a^	0.83 ^bc^	0.93 ^abc^	0.95 ^ab^	0.83 ^c^	0.87 ^bc^	0.91 ^abc^	0.02	0.006	0.001	0.32
C22:1	0.30 ^a^	0.24 ^ab^	0.23 ^b^	0.26	0.26	0.25	0.32 ^ab^	0.34 ^a^	0.24 ^ab^	0.22 ^ab^	0.24 ^ab^	0.27 ^ab^	0.23 ^ab^	0.21 ^b^	0.24 ^ab^	0.03	0.004	0.87	0.04
C20:5 n3[EPA]	8.03	8.04	7.99	7.69 ^b^	8.09 ^a^	8.27 ^a^	7.85 ^bcd^	7.97 ^abcd^	8.27 ^ab^	7.61 ^cd^	8.30 ^abc^	8.20 ^ab^	7.61 ^d^	8.00 ^abcd^	8.34 ^a^	0.09	0.76	0.001	0.08
C23:0	0.17 ^a^	0.11 ^b^	0.10 ^b^	0.16 ^b^	0.05 ^c^	0.17 ^a^	0.17 ^abc^	0.16 ^cd^	0.18 ^ab^	0.15 ^cd^	0.00 ^e^	0.18 ^a^	0.15 ^d^	0.00 ^e^	0.16 ^bcd^	0.00	0.001	0.001	0.001
C24:0	0.34	0.34	0.35	0.34	0.35	0.35	0.33 ^a^	0.36 ^a^	0.34 ^a^	0.33 ^a^	0.35 ^a^	0.34 ^a^	0.34 ^a^	0.34 ^a^	0.36 ^a^	0.01	0.25	0.12	0.05
C24:1	0.16 ^a^	0.08 ^b^	0.08 ^b^	0.13 ^a^	0.06 ^b^	0.13 ^a^	0.15 ^a^	0.17 ^a^	0.15 ^a^	0.11 ^ab^	0.00 ^b^	0.13 ^a^	0.13 ^a^	0.00 ^b^	0.11 ^a^	0.02	0.001	0.001	0.001
C22:5 n3[DPA]	2.57	2.57	2.57	2.54 ^b^	2.49 ^b^	2.68 ^a^	2.60 ^abc^	2.42 ^d^	2.70 ^a^	2.51 ^abcd^	2.56 ^abcd^	2.66 ^ab^	2.51 ^bcd^	2.50 ^cd^	2.70 ^a^	0.03	0.99	0.001	0.04
C22:6 n3[DHA]	34.4 ^b^	34.9 ^ab^	35.1 ^a^	34.0 ^b^	34.6 ^b^	35.8 ^a^	34.1 ^cd^	33.8 ^d^	35.4 ^abc^	33.8 ^bcd^	35.3 ^abcd^	35.7 ^ab^	34.2 ^bcd^	34.8 ^abcd^	36.2 ^a^	0.36	0.06	0.001	0.28
Unknown	3.64 ^a^	2.86 ^b^	2.94 ^b^	3.17 ^ab^	2.66 ^b^	3.60 ^a^	3.47 ^ab^	3.56 ^a^	3.88 ^a^	3.04 ^abc^	2.10 ^bc^	3.44 ^ab^	3.01 ^abc^	2.33 ^c^	3.48 ^a^	0.22	0.001	0.001	0.14
MUFA	20.2	20.3	19.8	20.2 ^a^	20.6 ^a^	19.4 ^b^	20.2 ^abc^	21.0 ^a^	19.3 ^c^	20.3 ^abc^	20.7 ^ab^	19.7 ^bc^	20.0 ^abc^	20.1 ^abc^	19.2 ^c^	0.23	0.04	0.001	0.42
PUFA	50.7	50.3	50.9	49.5 ^b^	50.0 ^b^	52.4 ^a^	50.1 ^bc^	49.8 ^bc^	52.1 ^ab^	48.9 ^c^	49.8 ^bc^	52.2 ^ab^	49.5 ^c^	50.3 ^bc^	52.9 ^a^	0.50	0.34	0.001	0.59
SFA	25.9 ^b^	26.4 ^ab^	26.5 ^a^	27.2 ^a^	27.0 ^a^	24.6 ^b^	26.3 ^b^	26.9 ^ab^	24.7 ^c^	27.8 ^a^	26.7 ^ab^	24.7 ^c^	27.5 ^a^	27.3 ^ab^	24.6 ^c^	0.24	0.02	0.001	0.01
n-3	47.4	47.2	48.0	46.4 ^b^	47.0 ^b^	49.2 ^a^	46.8 ^bc^	46.5 ^bc^	48.8 ^abc^	45.9 ^c^	46.8 ^bc^	48.9 ^ab^	46.6 ^bc^	47.7 ^abc^	49.8 ^a^	0.55	0.18	0.001	0.69
n-6	3.31 ^a^	3.07 ^b^	2.91 ^b^	3.08 ^ab^	2.97 ^b^	3.24 ^a^	3.32 ^a^	3.29 ^a^	3.32 ^a^	2.95 ^ab^	2.99 ^ab^	3.27 ^a^	2.96 ^ab^	2.64 ^b^	3.12 ^a^	0.08	0.001	0.004	0.07
n6/n3	0.07 ^a^	0.07 ^ab^	0.06 ^b^	0.07	0.06	0.07	0.07 ^a^	0.07 ^a^	0.07 ^a^	0.06 ^ab^	0.06 ^ab^	0.07 ^a^	0.06 ^ab^	0.06 ^b^	0.06 ^ab^	0.00	0.001	0.28	0.20

Values are presented as % of total fatty acids. PEF was conducted at different frequencies (25, 50, and 100 Hz) and intensities (0.625, 1.25, and 1.875 kV/cm). Different superscript letters for frequency, PEF field strength, and interaction (PEF frequency*PEF field strength) indicate a significant difference (*p* < 0.05). Means that are not statistically different within each factor or their interaction do not have any superscripts. Abbreviations: MUFA = monounsaturated fatty acid; PUFA = polyunsaturated fatty acid; SFA = saturated fatty acid, EPA = eicosapentaenoic acid; DPA = docosapentaenoic acid; DHA = docosahexaenoic acid; SEM = standard error mean.

**Table 6 foods-11-00610-t006:** Phospholipid composition of HMG pre-treated with PEF.

PL	Freq			PEF			Interaction									SEM		P	
	25	50	100	0.625	1.25	1.875	25 × 0.625	25 × 1.25	25 × 1.875	50 × 0.625	50 × 1.25	50 × 1.875	100 × 0.625	100 × 1.25	100 × 1.875		Freq	PEF	FRE*PEF
LDPG	0.27	0.35	0.35	0.27	0.35	0.35	0.16 ^b^	0.29 ^ab^	0.37 ^ab^	0.33 ^ab^	0.39 ^a^	0.33 ^ab^	0.33 ^ab^	0.38 ^a^	0.34 ^ab^	0.04	0.06	0.06	0.15
PG	0.13	0.13	0.13	0.15	0.13	0.12	0.16	0.12	0.11	0.14	0.13	0.13	0.13	0.14	0.12	0.01	0.99	0.07	0.28
CL	0.85	0.81	0.79	0.85	0.84	0.76	0.99	0.79	0.79	0.79	0.85	0.80	0.77	0.89	0.69	0.08	0.61	0.41	0.32
LPE	0.10	0.09	0.09	0.10	0.09	0.09	0.11	0.09	0.09	0.09	0.10	0.10	0.09	0.09	0.08	0.01	0.44	0.92	0.59
LPSP	0.50	0.48	0.45	0.49	0.49	0.45	0.58	0.44	0.47	0.47	0.51	0.47	0.42 ^a^	0.5	0.41	0.04	0.42	0.41	0.07
LPS	0.29 ^b^	0.37 ^ab^	0.40 ^a^	0.28 ^b^	0.40 ^a^	0.38 ^a^	0.18 ^b^	0.33 ^ab^	0.38 ^ab^	0.28 ^ab^	0.46 ^a^	0.36 ^ab^	0.37 ^ab^	0.42 ^a^	0.41 ^a^	0.04	0.03	0.01	0.30
SM	0.38	0.45	0.36	0.40	0.43	0.37	0.37	0.39	0.39	0.48	0.48	0.39	0.34	0.41	0.33	0.07	0.30	0.61	0.94
PE	3.75	3.45	3.22	3.81	3.31	3.30	4.65 ^a^	3.32 ^ab^	3.29 ^ab^	3.37 ^ab^	3.35 ^ab^	3.62 ^ab^	3.43 ^ab^	3.26 ^ab^	2.98 ^b^	0.33	0.20	0.13	0.17
LPC-1	0.53	0.49	0.49	0.51	0.52 ^a^	0.4	0.60	0.48	0.51	0.45	0.53	0.49	0.49	0.56	0.41	0.05	0.57	0.47	0.27
PS	0.78	0.74	0.72	0.80	0.75	0.69	0.94	0.72	0.67	0.73	0.75	0.74	0.74	0.76	0.66	0.07	0.64	0.19	0.26
LPC-2	0.09	0.07	0.07	0.09	0.08	0.07	0.12	0.08	0.07	0.06	0.09	0.07	0.07	0.08	0.07	0.02	0.36	0.38	0.20
PI	0.13	0.12	0.12	0.13	0.12	0.12	0.15	0.09	0.09	0.12	0.12	0.13	0.11	0.14	0.14	0.02	0.58	0.90	0.20
PC	8.14	8.60	8.02	8.00	8.69	8.07	8.86	7.99	7.57	7.58	9.32	8.90	7.55	8.77	7.75	0.70	0.58	0.43	0.33

Values are presented as µmol/g wet tissue. PEF was conducted at different frequencies (25, 50, and 100 Hz) and intensities (0.625, 1.25, and 1.875 kV/cm). Data were obtained by ^31^P NMR. Means for each factor separately and for the interaction of the factors, which do not share a superscript letter, are not significantly different (*p* > 0.05, Tukey’s test). Means that are not statistically different within each factor or their interaction do not have any superscripts. Abbreviation: PA = phosphatidic acid; LDPG = lyso-diphosphatidylglycerol; CL = cardiolipin; LPE = lyso-phosphatidylethanolamine; LPS = lyso-phosphatidylserine; SM = sphingomyelin; PE = phosphatidylethanolamine; LPC = lyso-phosphatidylcholine; PS = phosphatidylserine; PI = phosphatidylinositol; PC = phosphatidylcholine, SEM = standard error mean.

## Data Availability

The data presented in this study are available on request from the corresponding author.

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
