# Peer review of "Effect of Pulsed Electric Fields on the Lipidomic Profile of Lipid Extracted from Hoki Fish Male Gonad"

_foods, 2022, doi:10.3390/foods11040610_

Round 1
Reviewer 1 Report
Reviewer comments:
The current study was to investigate the use of pulsed electric fields (PEF) as a pre-treatment for enhancing the extraction of lipid extracted from hoki male gonad. The paper is supported by relevant experimental data, the presentation is clear and most of the conclusions are justified. However, a major revision needs to be response.
Line 20: The abbreviation “PEF” appears for the first time, please indicate the full form.
Line 49: A comma or semicolon is missing after the “moisture: 72-73%”.
Line 229: “...the highest temperature (52.8ºC) was found at the highest PEF field strength of 1.875 kV and frequency of 100 Hz that was used in the study.”, however, in Table 2, the temperature under the above conditions is 45.0ºC, which is inconsistent with the description. Please give a reasonable explanation.
Line 234: Change “(P < 0.05)” to “(P < 0.05)”, the similar problems should be corrected.
Line 239-240: “However, the use of PEF as a pre-treatment was not found to be effective with PEF pre-treatment under the other conditions, …”, this sentence is too wordy, please rephrase it.
Line 296: “In addition, there was a decrease in some SFA (C24:0), MUFA (C14:1 and C17:1) and PUFA (C18:3 n6 and C20:3 n3), as a result of the heating effect.”, it would be better if the authors could explain in detail the relationship between “heating effect” and the loss of these fatty acids.
Line 317: In Table 4, some “P” values are in bold, please unify the format.
Line 329: In this section, the authors mainly described the changes of fatty acid composition under PEF treatment. I think the relevant explanations need to be added in this part. Similar problems are also found in other sections, please check the whole paper.
Line 496-501: In conclusions, the statement of “Previous literature has demonstrated ……” is inappropriate for this section. I think it is probably more suitable for the part of introduction. The conclusions should be rewritten.
Author Response
The authors would like to thank the Editor for handling our manuscript, and the Reviewers for their comments on our manuscript which have helped us improve the overall quality of our manuscript.
Reviewer #1
The current study was to investigate the use of pulsed electric fields (PEF) as a pre-treatment for enhancing the extraction of lipid extracted from hoki male gonad. The paper is supported by relevant experimental data, the presentation is clear and most of the conclusions are justified. However, a major revision needs to be response.
Response: We thank Reviewer for their positive comment.
Line 20: The abbreviation “PEF” appears for the first time, please indicate the full form.
Response: Done.
Line 49: A comma or semicolon is missing after the “moisture: 72-73%”.
Response: Done.
Line 229: “...the highest temperature (52.8ºC) was found at the highest PEF field strength of 1.875 kV and frequency of 100 Hz that was used in the study.”, however, in Table 2, the temperature under the above conditions is 45.0ºC, which is inconsistent with the description. Please give a reasonable explanation.
Response: We thank Reviewer for spotting this oversight. The ‘highest temperature’ has been corrected to 45.0ºC.
Line 234: Change “(P < 0.05)” to “(P < 0.05)”, the similar problems should be corrected.
Response: Reviewer 3 requested that the ‘P’ in the expression should be lower case and italic. We checked a recent publication in Foods journal (https://www.mdpi.com/2304-8158/11/3/374/htm) and found that lower case and italic was used. Therefore, to be consistent, we have elected to use lowercase and italic for the letter P.
Line 239-240: “However, the use of PEF as a pre-treatment was not found to be effective with PEF pre-treatment under the other conditions, …”, this sentence is too wordy, please rephrase it.
Response: The sentence has been re-phrased to meet the Reviewer’s comment, as follows:
“However, the use of PEF as a pre-treatment did not result in a significant (p > 0.05) difference in total lipid yield compared to that obtained from untreated HMG.”
Line 296: “In addition, there was a decrease in some SFA (C24:0), MUFA (C14:1 and C17:1) and PUFA (C18:3 n6 and C20:3 n3), as a result of the heating effect.”, it would be better if the authors could explain in detail the relationship between “heating effect” and the loss of these fatty acids.
Response: To meet the Reviewer’s comment, we have modified the discussion and supported the text with relevant references.
“The long chain saturated and unsaturated fatty acids (both MUFA and PUFA) could be decomposed and oxidized at high temperature and could convert into secondary oxidative products [33][34][35], resulting in reduction of the fatty acid content. This phenomenon might explain the apparent decrease in the amount of SFA (C24:0), MUFA (C14:1 and C17:1) and PUFA (C18:3 n6 and C20:3 n3) in the heat-treated control.”
Line 317: In Table 4, some “P” values are in bold, please unify the format
Response: All ’P’ values have been made consistent (lower case and italic) as per the comment above.
Line 329: In this section, the authors mainly described the changes of fatty acid composition under PEF treatment. I think the relevant explanations need to be added in this part. Similar problems are also found in other sections, please check the whole paper.
Done. The following sentences have been added.
“It is well known that PEF treatment causes electroporation of cell membranes [xx] and facilitate the release of cellular contents. The above results suggests that different fatty acids respond differently to the PEF treatment intensity.”
“As mentioned earlier, the electroporation caused by PEF could enhance the release of cellular contents. However, with the increase in treatment intensity, heat generated due to joule effect appear to degrade some compounds such as PE.”
Line 496-501: In conclusions, the statement of “Previous literature has demonstrated ……” is inappropriate for this section. I think it is probably more suitable for the part of introduction. The conclusions should be rewritten.
Response: We agree with the Reviewer and have moved the sentence to the introduction as suggested. We have also reviewed the text in the Conclusions section and have improved the clarity.
Reviewer 2 Report
The manuscript “Effect of pulsed electric fields on the lipidomic profile of lipid extracted from hoki fish male gonad” describes the extraction of lipids from hoki fish male gonad using a modern Pulsed Electric Field (PEF) aided technology. The authors applied the pulse of different strengths and frequencies to determine the effects on lipidomic profile; additionally total lipid was extracted by ethanol-hexane based (ETHEX) extraction method. Moreover, the extraction yield and lipid profile of the oil extracted by those methods were compared. This is an interesting study where the authors reported the findings of lipid profiles affected by the extraction conditions. The manuscript is well written, well-balanced, and nicely described. The scientific value and language is sound. The approach and study used in this study is novel and the contents of this paper fit the scope of this journal. However, there are some minor issues need be solved.
Specific Comments:
- Abstract: Please write the elaborative form of PEF in first use.
- Introduction: The authors mentioned that the gonad and liver of hoki could comprise 10-25% of the total fish. Could the authors add any information about solely the male gonad yield (% of total fish weight)?
- Table 1: What does HG denote for? If it is hoki gonad, then I prefer to express as hoki male gonad (HMG) and the authors should write the elaborative form in first use. The abbreviated form could be used throughout the manuscript rather than only in Table 1.
- Table 3: The alphabetic letters denoting significant changes in values should be presented as superscript.
- Table 5: There are some values on which there is no superscript letters. E.g., values of C21:0, C20:3 n3, C20:5 n3 [EPA] regarding PEF frequency. Why did the authors exclude to compare the significant changes of those values?
- Table 5: Please mention the elaboration of SEM. I think the individual Table should be self-explanatory.
- Table 6. Turkey’s test should be Tukey’s test.
- Conclusion: The authors stated “Very little difference was detected in hoki male gonad samples pre-treated with PEF in terms of the extracted lipid. The phospholipid composition was not substantially changes”. Additionally, the n-3 fatty acids content in total oil extracted by ETHEX method and PEF aided method seem very closer (Table: 3 and Table: 5). It is difficult for the readers to understand the advantages of PEF treated extraction.
Author Response
The authors would like to thank the Editor for handling our manuscript, and the Reviewers for their comments on our manuscript which have helped us improve the overall quality of our manuscript.
Reviewer #2
The manuscript “Effect of pulsed electric fields on the lipidomic profile of lipid extracted from hoki fish male gonad” describes the extraction of lipids from hoki fish male gonad using a modern Pulsed Electric Field (PEF) aided technology. The authors applied the pulse of different strengths and frequencies to determine the effects on lipidomic profile; additionally total lipid was extracted by ethanol-hexane based (ETHEX) extraction method. Moreover, the extraction yield and lipid profile of the oil extracted by those methods were compared. This is an interesting study where the authors reported the findings of lipid profiles affected by the extraction conditions. The manuscript is well written, well-balanced, and nicely described. The scientific value and language is sound. The approach and study used in this study is novel and the contents of this paper fit the scope of this journal. However, there are some minor issues need be solved.
The authors thank the Reviewer for their positive comments about our manuscript.
Specific Comments:
- Abstract: Please write the elaborative form of PEF in first use.
Response: Done.
- Introduction: The authors mentioned that the gonad and liver of hoki could comprise 10-25% of the total fish. Could the authors add any information about solely the male gonad yield (% of total fish weight)?
Response: The authors do not have the weight proportion of the male gonad available, as the authors did not process the fish and the information is not available from the fish processor. To the best of our knowledge there are no reports in the literature on this aspect for hoki.
- Table 1: What does HG denote for? If it is hoki gonad, then I prefer to express as hoki male gonad (HMG) and the authors should write the elaborative form in first use. The abbreviated form could be used throughout the manuscript rather than only in Table 1.
Response: This has been corrected to ‘hoki male gonad’ and included in the text initially in the elaborative form, followed subsequently by the abbreviated form ‘HMG’, throughout the manuscript.
- Table 3: The alphabetic letters denoting significant changes in values should be presented as superscript.
Response: Done.
- Table 5: There are some values on which there is no superscript letters. E.g., values of C21:0, C20:3 n3, C20:5 n3 [EPA] regarding PEF frequency. Why did the authors exclude to compare the significant changes of those values?
Response: Statistical analysis was conducted on all of the data presented in the manuscript. We are of the opinion that where there is no significant variation (p > 0.05) within the treatment, there is no requirement for a superscript letter. We have added a footnote to the tables to clarify this point.
- Table 5: Please mention the elaboration of SEM. I think the individual Table should be self-explanatory.
Response: We have added the elaboration of ‘SEM’ in the abbreviation section of the manuscript to meet the Reviewer’s comment.
- Table 6. Turkey’s test should be Tukey’s test.
Response: This has been corrected
- Conclusion: The authors stated “Very little difference was detected in hoki male gonad samples pre-treated with PEF in terms of the extracted lipid. The phospholipid composition was not substantially changes”. Additionally, the n-3 fatty acids content in total oil extracted by ETHEX method and PEF aided method seem very closer (Table: 3 and Table: 5). It is difficult for the readers to understand the advantages of PEF treated extraction.
Response: We have reviewed the Conclusions section and have improved the text.
Reviewer 3 Report
The present study focused on analysing the effect of pulsed electric fields on the extracted lipid yield and lipid profile of hoki male gonad. The results revealed that pulsed electric fields pre-treatment under mild conditions has a potential for improving the total lipid yield extracted from fish gonad.
Generally, the manuscript is a great piece which was properly written. The research was appropriately conceptualized. The Abstract was nicely written; was informative enough and provided sufficient information on the objectives of the study, methodology, results and conclusions made from the study. The Introduction was also great. The authors provide a good funnel-shaped introduction that highlighted the background of study and led to the exposure of the knowledge gap that needed to be filled with the present study. The methods and procedures followed for the whole experiment were adequately and properly provided, in addition to the statistical methods applied for the analysis of the obtained results. The results and discussions sections were nicely done also, with a great level of scientific tone and discussive coherence.
Minor comments:
Could you add a simple back ground in the beginning of the abstract.
In the statistical analysis did the authors examine if the data meets the parametric test before the Two Way ANOVA.
The (P < 0.05) could be written in a lower case and italicized .
The reference style is not correct revised it along the manuscript.
Author Response
The present study focused on analysing the effect of pulsed electric fields on the extracted lipid yield and lipid profile of hoki male gonad. The results revealed that pulsed electric fields pre-treatment under mild conditions has a potential for improving the total lipid yield extracted from fish gonad.
Generally, the manuscript is a great piece which was properly written. The research was appropriately conceptualized. The Abstract was nicely written; was informative enough and provided sufficient information on the objectives of the study, methodology, results and conclusions made from the study. The Introduction was also great. The authors provide a good funnel-shaped introduction that highlighted the background of study and led to the exposure of the knowledge gap that needed to be filled with the present study. The methods and procedures followed for the whole experiment were adequately and properly provided, in addition to the statistical methods applied for the analysis of the obtained results. The results and discussions sections were nicely done also, with a great level of scientific tone and discussive coherence.
We thank the Reviewer for their positive comments on our manuscript.
Minor comments:
Could you add a simple back ground in the beginning of the abstract.
Response: a brief background has been added to the beginning of the abstract.
In the statistical analysis did the authors examine if the data meets the parametric test before the Two Way ANOVA.
Response: Yes, a normality test and a homogeneity test were performed before the ANOVA. We have added the following text to clarify.
“The homogeneity of variance and normality of the data were determined using Bartlett’s test and Shapiro-Wilk test, respectively.”
The (P < 0.05) could be written in a lower case and italicized.
Response: Done.
The reference style is not correct revised it along the manuscript.
Response: The reference style has been revised throughout the manuscript.